# Emergence of *bla*_NDM-5_ and *bla*_OXA-232_ Positive Colistin- and Carbapenem-Resistant *Klebsiella pneumoniae* in a Bulgarian Hospital

**DOI:** 10.3390/antibiotics13070677

**Published:** 2024-07-21

**Authors:** Rumyana Markovska, Petya Stankova, Georgi Popivanov, Ivanka Gergova, Kalina Mihova, Ventsislav Mutafchiyski, Lyudmila Boyanova

**Affiliations:** 1Department of Medical Microbiology, Medical Faculty, Medical University of Sofia, 1431 Sofia, Bulgaria; petya.stankova88@abv.bg (P.S.); l.boyanova@hotmail.com (L.B.); 2Department of Surgery, Military Medical Academy, 1606 Sofia, Bulgaria; gerasimpopivanov@rocketmail.com (G.P.); ventzimm@gmail.com (V.M.); 3Department of Microbiology and Virology, Military Medical Academy, 1606 Sofia, Bulgaria; gergova.ivana7@gmail.com; 4Molecular Medicine Center, Medical University of Sofia, 1431 Sofia, Bulgaria; kmihova@mmcbg.org

**Keywords:** NDM-5, OXA-232, *K. pneumoniae*, Bulgaria

## Abstract

The rapid spread of carbapenemase-producing strains has led to increased levels of resistance among Gram-negative bacteria, especially enterobacteria. The current study aimed to collect and genetically characterize the colistin- and carbapenem-resistant isolates, obtained in one of the biggest hospitals (Military Medical Academy) in Sofia, Bulgaria. Clonal relatedness was detected by RAPD and MLST. Carbapenemases, ESBLs, and *mgrB* were investigated by PCR amplification and sequencing, replicon typing, and 16S rRNA methyltransferases with PCRs. Fourteen colistin- and carbapenem-resistant *K. pneumoniae* isolates were detected over five months. Six carbapenem-resistant and colistin-susceptible isolates were also included. The current work revealed a complete change in the spectrum of carbapenemases in Bulgaria. *bla*_NDM-5_ was the only NDM variant, and it was always combined with *bla*_OXA-232_. The coexistence of *bla*_OXA-232_ and *bla*_NDM-5_ was observed in 10/14 (72%) of colistin- and carbapenem-resistant *K. pneumoniae* isolates and three colistin-susceptible isolates. All *bla*_NDM-5_- and *bla*_OXA-232_-positive isolates belonged to the ST6260 (ST101-like) MLST type. They showed great *mgrB* variability and had a higher mortality rate. In addition, we observed *bla*_OXA-232_ ST14 isolates and KPC-2-producing ST101, ST16, and ST258 isolates. The colistin- and carbapenem-resistant isolates were susceptible only to cefiderocol for *bla*_NDM-5_- and *bla*_OXA-232_-positive isolates and to cefiderocol and ceftazidime/avibactam for *bla*_OXA-232_- or *bla*_KPC-2_-positive isolates. All *bla*_OXA-232_-positive isolates carried *rmtB* methylase and the colE replicon type. The extremely limited choice of appropriate treatment for patients infected with such isolates and their faster distribution highlight the need for urgent measures to control this situation.

## 1. Introduction

*Klebsiella pneumoniae* is an important nosocomial pathogen. It can cause a wide range of nosocomial infections such as the bloodstream, surgical wounds, urinary infections, pneumonia, and catheter-associated infections [1]. It also causes necrotizing pneumonia, pyogenic liver abscesses, and endogenous endophthalmitis [1]. In the last years, the resistance of isolates from this species has increased mostly due to the rising frequency of extended-spectrum beta-lactamase production and carbapenemase carriers. The most widespread carbapenemases are class A *Klebsiella*-producing carbapenemase (KPC), class D oxacillinases (OXA)—OXA-48, OXA-181, and their variants—as well as the class B metallo-beta-lactamases (MBLs) (New Delhi Metalo beta-lactamase, NDM; imipenemase, IMP; and Verona Integron Metallo-carbapenemase, VIM) [2]. NDM-, KPC- and OXA-48-producing enterobacteria, especially *Klebsiella pneumoniae*, are widely distributed all over the world, involving many international high-risk clones as ST258, ST11, ST15, ST395, and causing serious therapeutic problems [2,3,4]. KPC enzymes are endemic in the USA and some European countries such as Greece and Italy [5]. OXA-48 and its variants were widely detected in Turkey and some European countries, such as Spain and France; particularly, OXA-232 is a new, emerging, and quite successful variant in Asia and Turkey [2,3,4,5,6]. NDM-producing enterobacteria appeared for the first time in India and was then distributed all over the world [2,5]. The carbapenemase producers are widely resistant to almost all beta-lactams and many other antibiotics such as colistin (polymixin E), aminoglycosides, quinolones, and tetracyclines [3]. Some isolates are extremely resistant or pan-resistant, which is a serious problem for the therapy [3].

The rapid spread of carbapenemase-producing strains has led to renewal of polymyxin use in medicine. The increased and inappropriate colistin (polymyxin E) use was followed, however, by a faster appearance of lipopolysaccharide (LPS) modifications: an addition of 4-amino-4-deoxi-L-arabinose or phosphoethanolamine. These modifications occur due to mutations in *pmrCAB* and *arnBCADTEF-pmrE* operons and are associated with an increase in the positive charge in LPSs and, therefore, a decrease in colistin binding [7]. They are under the control of the two-component systems, PhoP/PhoQ and PmrA/PmrB. The PhoP/PhoQ system is subjected to negative-feedback regulation by MgrB protein [7,8]. In recent years, plasmid-mediated colistin resistance encoded by *mcr* genes has also been reported [9]. The *mcr* genes encoding phosphoethanolamine transferase add phosphoethanolamine and diminish the negative charge of LPS, thus decreasing the affinity of LPS to colistin.

In addition to the beta-lactams and colistin, another important antibiotic group is that of aminoglycosides. They could be used in combination with beta-lactams or other antibiotics, especially for treatment of infections caused by carbapenemase-producing enterobacteria [10]. The main mechanism of resistance to them is the production of aminoglycoside-modifying enzymes (phosphotransferases, adenylyltransferases, nucleotidyltransferases, and acetyltransferases). Each of those enzymes may confer resistance to some aminoglycosides but not to all of them. In the last few decades, 16S rRNA methyltransferase enzymes (16S rRNTases) have increasingly been detected. They methylate 16S rRNA subunit; thus, the aminoglycosides cannot bind to the ribosome, which confer high-level resistance to all clinically relevant aminoglycosides. Many variants of RTNases have been detected (armA, rmtA-H, and NpmA) [11]. They are plasmid-encoded, have the possibility to be easily transferred and have often been found in carbapenemase-producing isolates, which, taking into account other mechanisms, can lead to extremely resistant or pandrug-resistant Gram-negative isolates [10,11,12].

In Bulgaria, carbapenemases from the four main groups were detected and NDM (NDM-1) and KPC (KPC-2) were predominant [13,14,15], but VIM-1 and OXA-48 [14,16,17] were also found. The frequency of colistin-resistant isolates also increased [18,19,20].

This study aimed to collect and genetically characterize colistin- and carbapenem-extremely resistant isolates in one of the biggest hospitals (Military Medical Academy) in Sofia, Bulgaria.

## 2. Materials and Methods

### 2.1. Bacterial Isolates and Patients’ Susceptibility Testing

Colistin- and carbapenem-resistant *Enterobacterales* isolates were collected in the Military Medical Academy, Sofia, Bulgaria, from September 2023 to January 2024. Such isolates started to appear in August 2023 [20], and the last isolate was detected at the end of January 2024. Data about the patients such as the month of isolation, sample, ward, prior treatment, and diagnosis are presented in Table 1. Only one isolate per patient was included in this study when the isolates were identical. Routine methods were used to carry out the cultivation. The samples were obtained during routine therapeutic and diagnostic work. The project was approved by the Medical Science Council of Medical University, Sofia.

Identification and susceptibility testing to the antibiotics (amoxicillin, amoxicillin/clavulanic acid, piperacillin/tazobactam, cefuroxime, ceftriaxone, ceftazidime, cefepime, imipenem, meropenem, colistin, amikacin, gentamicin, fosfomycin, levofloxacin, ciprofloxacin, and trimethoprim/sulfamethoxazole) were carried out via Vitek 2 (bioMerieux, Marcy-l'Étoile, France).

The disk diffusion method (30 µg disk, Liofilchem, Roseto degli Abruzii, Italy) was used for ceftazidime/avibactam susceptibility testing. Minimal inhibitory concentrations (MICs) of eravacycline and plazomicin were detected by E-tests (Liofilchem, Roseto degli Abruzii, Italy), and cefiderocol and colistin susceptibility was determined by a broth microdilution method (ComASP Cefideracol with iron-depleted broth, Liofilchem, Italy and ComASP Colistin, Liofilchem, Italy, respectively). The analysis of the results was based on the breakpoints of the European Committee on Antimicrobial Susceptibility Testing (EUCAST), 2024 [21] and the Food and Drug Administration (FDA) for plazomicin and eravacycline [22,23]. FDA breakpoints of ≤0.5 mg/L for susceptibility and ≥1 mg/L for non-susceptibility were considered for eravacycline [22]. For plazomicin, the manufacturers/FDA breakpoints were ≤2 mg/L for susceptibility, 4 mg/L for intermediate susceptibility, and ≥8 mg/L for resistance [23]. Multidrug resistance (MDR) was defined according to their non-susceptibility to at least one agent in ≥3 antibiotic classes. Extensive drug resistance (XDR) was found when isolates were non-susceptible to one or more agents in all but two or fewer antimicrobial categories [24].

### 2.2. Molecular Identification of Beta-Lactamases and 16S rRNA Methylases

The detection of *bla*_VIM_, *bla*_IMP_, *bla*_KPC_, *bla*_NDM_, *bla*_OXA-48_
*bla*_CTX-M_, *bla*_SHV_, *bla*_CMY_, *bla*_FOX_, *bla*_DNA_, and *bla*_ACC_ genes was performed as previously described [25,26]. The genes were amplified and sequenced using primers as previously described *bla*_CTX-M-1-group_, *bla*_SHV,_
*bla*_KPC_ [14], *bla*_NDM_ [27], and, for the *bla*_OXA-48_ primer set, OXA-F [28] and OXA-R [29]. Nucleotide sequences were analysed using Chromas Lite 2.01 (Technelysium Pty Ltd., Brisbane, Australia) and DNAMAN version 8.0 Software (Lynnon BioSoft, Vaudreuil-Dorion, QC, Canada). The primers and protocol for the detection of *armA*, *rmtB*, and *rmtC* were considered as previously described [30].

### 2.3. Molecular Typing

Whole-cell DNA was prepared by a DNA Swab Isolation Spin Kit (Illustra Bacteria Genomic DNA Prep Kit (GE Healthcare, Little Chalfont, UK). ERIC (Enterobacterial Repetitive Intergenic Consensus) PCR was performed with ERIC 1 and ERIC 2A primers as previously described [14]. ERIC fingerprints were used to observed genetic similarity using the simple clustering method, UPGMA (unweighted-pair group method with arithmetic mean), and the Dice coefficient as a similarity measure (http://genomes.urv.cat/UPGMA/ accessed on 1 June 2024). A clone was defined when the coefficient of similarity of the isolates was above 0.8 (http://genomes.urv.cat/UPGMA/ accessed on 1 June 2024).

For *K. pneumoniae* species complex isolates, multilocus sequence typing (MLST) including seven conserved housekeeping genes (*gapA*, *infB*, *mdh*, *pgi*, *phoE*, *rpoB*, and *tonB*) was made according to https://bigsdb.pasteur.fr/klebsiella/primers-used/ accessed on 1 June 2024. Determination of the allelic numbers and sequence types (STs) was according to the MLST database (Pasteur Institute, Paris, France; https://bigsdb.pasteur.fr/klebsiella/ accessed on 1 June 2024).

### 2.4. Replicon Typing

We performed replicon typing by the PCR-based replicon typing scheme described by Carattoli et al. [31] using whole-cell DNAs extracted from the isolates.

### 2.5. mgrB Sequencing and Plasmid Colistin Resistance

We investigated the plasmid-mediated colistin resistance determinants (mcr-1, mcr-2, mcr-3, mcr-4, and mcr-5) via multiplex PCR as described previously [32]. The sequencing of *mgrB* was carried out according to Cannatelli et al., 2014 [8]

## 3. Results

### 3.1. Bacterial Isolates and Patients’ Susceptibility Testing

From September 2023 to January 2024, we collected 14 colistin-resistant and carbapenem-non-susceptible *Klebsiella pneumoniae* in the Military Medical Academy, Sofia, Bulgaria. To obtain an insight into the distributed clones and enzymes, we included six randomly selected colistin-susceptible and carbapenem-resistant isolates of *K. pneumoniae*, detected in January 2024.

*K. pneumoniae* isolates included in the current study were detected from patients hospitalized in intensive care units or surgery wards and were predominantly from blood cultures (50% of isolates). The source of other isolates (tracheo-bronchial secretions, n = 3; drain fluids, n = 3; pleural punctate, n = 1; abscess, n = 1; and wound, n = 1) is shown on Table 1. Fifteen isolates (75%) were obtained from men. The median age of the patients was 68.4 years (40–91 years). Prevalent diagnosis included different types of cancer (8 patients), peritonitis (2), and pyelonephritis/kidney insufficiency (5) (Table 1). Eleven patients had fatal outcomes: nine of them (64%, 9/14) from the colistin-resistant group and two (33%, 2/6) from the colistin-susceptible group.

The susceptibility of isolates is shown in Table 2. The colistin- and carbapenem-resistant isolates were also resistant to all the antimicrobials tested, which makes them pandrug-resistant according to Magiorakos’ criteria [24]. Additional testing found that only cefideracole susceptibility of the isolates was 100%, while the susceptibility to ceftazidime/avibactam was 35% (7/20) (four colistin-resistant and three colistin-susceptible isolates were susceptible to ceftazidime/avibactam). The isolates were highly resistant to eravacycline and plazomicin.

### 3.2. Molecular Identification of Beta-Lactamases and 16S rRNA Methylases

PCR and sequencing revealed the presence of three carbapenemase genes (*bla*_OXA- 232_, *bla*_NDM-5_ and *bla*_KPC-2_) (Table 1). *bla*_OXA-232_ was detected in 80% (16 isolates, 13 of them colistin-resistant and 3 colistin-susceptible). Twelve of the *bla*_OXA- 232_-positive isolates were also positive for *bla*_NDM-5_, and *bla*_KPC-2_ was detected in one colistin-resistant and three colistin-susceptible isolates. From the CTX-M family, only four group isolates positive for *bla*_CTX-M-1_ were detected (Table 1). The sequence revealed the presence of *bla*_CTX-M-15_.

All isolates were positive for the SHV group-specific primers set and almost all showed a presence of *bla*_SHV-28_ genes, except two isolates that produced KPC enzymes, and the SHV variant in these isolates was *bla*_SHV-1_.

Among the 17 isolates highly resistant to aminoglycosides, 12 showed the presence of *rmtB* and three had the presence of *armA* (Table 1).

### 3.3. Molecular Typing

ERIC PCR produced 8–12 bands, and according to the fingerprint UPGMA analysis, it revealed the presence of five clones with a similarity below 0.8; A and A’ have a similarity coefficient of 0.92 which combines them into one clone. Five MLST types were observed. The predominant was ST6260: it contained 13 *bla*_OXA- 232_- or *bla*_NDM-5_-positive isolates (including 10 (71.4%) colistin-resistant and 3 colistin-susceptible isolates). This ST type belonged to CC11 and was the closest to ST101 (one allele (pho) difference with ST101) (Table 1). The other colistin-resistant ST types were ST16 (one member, *bla*_KPC-2_- and *bla*_CTX-M-15_-positive), ST 14 (three members, *bla*_OXA- 232_- and *bla*_CTX-M-15_-positive), and the other colistin-susceptible types were ST258 (one isolate, *bla*_KPC-2_- and *bla*_CTX-M-15_-positive), ST101 (two isolates, *bla*_KPC-2_-positive).

### 3.4. Replicon Typing

Sixteen isolates were positive for the colE replicon type. They included all *bla*_OXA-232_-positive isolates with or without NDM-5 production. KPC-2-producing isolates showed a presence of plasmids of IncFII_AS_ type (Table 1).

### 3.5. mgrB and Plasmid Colistin Resistance

All isolates were negative for *mcr*-1, -2, -3, -4, and -5. Among the 14 colistin-resistant isolates, a disruption in the *mgrB* locus with a lack of amplification was found in six, and sequencing revealed an exchange in position a124t in three isolates. This resulted in an exchange of asparagin N with tyrosin Y in position 42. One isolate had a 17 bp deletion from 72 to 89 nucleotides causing frameshift and premature termination. Four isolates displayed wild types.

## 4. Discussion

The number of infections caused by multidrug-resistant bacteria has been increasing worldwide. Antibiotic resistance has been considered one of the biggest threats to human health [3]. The frequency of carbapenemase-producing *K. pneumoniae* in Bulgaria has increased from 7.2% in 2018 to 19.3% in 2021 (https://www.bam-bg.net/index.php/bg/bulstar accessed on 15 June 2024). This percentage is nearly twice as high as those for Europe: 10.9% for 2022 [33]. This sharp increase in carbapenem-resistant isolates in the country reflects the increase in invasive *K pneumoniae* isolates. In Europe, the frequency of carbapenem-resistant invasive *K. pneumoniae* isolates rose sharply (by 49.7%) from 2019 to 2022 [33]. The same trend was observed in Bulgaria, where the frequency of these isolates increased from 0% in 2013 to 21.2% in 2018 and 47.3% in 2022, with the biggest increase during the last two years [33]. This coincides with the huge increase in total antibiotic consumption in Bulgaria during the last few years. Bulgaria was in third place in Europe in 2022 for community antibiotic usage with 25.2 DDD per 1000 inhabitants per day, and around 1.49 DDD per 1000 inhabitants per day (hospital usage) with a higher prevalence of cephalosporines and carbapenems [34].

The spectrum of carbapenemases among enterobacterial isolates has also changed over the years. VIM-1 was the first carbapenemase detected in the enterobacteria in Bulgaria (observed in *P. mirabilis*) [35]. During the following years, an increased frequency of KPC-2-producing *K. pneumoniae* was detected [13,14]. NDM-1 in combination with CTX-M-15/-3 and with or without CMY-4 has been predominant among the carbapenemase-producing *K pneumoniae* isolates since 2017 [13,36]. The situation was similar with faecal-carriage isolates: of 14 isolates proved as carbapenemase producers, 11 (79%) were *bla*_NDM-1_-positive (for hospitals) [37]. *bla*_NDM-1_ also prevailed among colistin- and carbapenem-resistant isolates in 2018/2019 [19], since among 37 colistin-resistant/heteroresistant and carbapenem-resistant isolates, we found 25 isolates (68%) carrying *bla*_NDM-1_.

Interestingly, the current work revealed a complete change in the spectrum of carbapenemases in Bulgaria. *bla*_NDM-5_ was the only one NDM variant in the current study, and in every case, it was combined with *bla*_OXA-232_.

NDM-5 was first identified in the ST648 *Escherichia coli* isolate in 2011 from a patient with a hospitalization history in India [38]. It differed from *bla*_NDM-1_ by two point mutations at positions 262 (G →T) and 460 (A →C) responsible for the two amino acid substitutions 88 (Val→Leu) and 154 (Met→Leu) [38]. Subsequently, this carbapenemase was detected in *K. pneumoniae*, and its producers caused sporadic outbreaks worldwide [39].

OXA-232 differs from OXA-181 by one amino acid (aa) substitution (Arg214Ser) and from OXA-48 by five aa substitutions [40]. OXA-232 has weaker carbapenem hydrolytic activity, but a stronger penicillin hydrolysing activity when compared with OXA-48 and OXA-181 [41]. This observation was proved by our results. We observed three isolates that produced only OXA-232 and had intermediate susceptibility to imipenem. This enzyme was initially described in France (in patients returning from India), and thereafter, it was detected in China, India, South Korea, Singapore, Thailand, and the USA [40,42,43,44].

The coexistence of *bla*_OXA-232_ and *bla*_NDM-5_ was observed for 10/14 (72%) of colistin-resistant and carbapenem-non-susceptible *K. pneumoniae* isolates and for three of colistin-susceptible isolates in the current study. For the first time, this combination has been detected in Nepal [45]. Similar reports have been published in India [46], Italy [47], and Bangladesh [48]. Coproduction of NDM-5 and OXA-232 has been reported for various spectrums of STs types: ST14, ST16, ST147, ST231, ST2497, and ST427 (CC258) [46,47]. Interestingly, the isolates found in the current study belonged to the rare ST type (ST6260), which differs from ST101 by only one allele, *pho*601 (ST6260), instead of *pho*4 (for ST101). Because of the difference of only one nucleotide between ST101 and ST6260, we hypothesize that ST101 is a possible progenitor of ST6260. To the best of our knowledge, this clone has not previously been reported to carry ESBLs or carbapenemase genes. ST101 is a well-known international clone, which has been reported to carry a wide range of carbapenemases such as OXA-48 [42]. It is not clear, however, whether the mutation in the pho allele occurred in the Military Medical Academy or another hospital. Further studies should be performed in more hospitals in Sofia and Bulgaria to reveal the development of this cluster. Ten colistin/carbapenem-resistant and three colistin-susceptible/carbapenem-resistant isolates belonged to the NDM-5- and OXA-232-producing ST6260 *K pneumoniae* cluster. All the isolates showed the presence of the ColE replicon type, which is in concordance with reports about the *bla*_OXA-232_ gene being usually located on a 6-kb nonconjugative ColE-type plasmid [40]. In addition, three of our isolates, positive only for *bla*_OXA-232_, showed the presence of the ColE-type plasmid. Except this plasmid type, *bla*_OXA-232_ was also located on ColK3 plasmid types, which showed a similarity to ColE plasmids [6,49].

The increased number of NDM-5 and OXA-232 producers can be associated with the high mobility of the encoding genes, their genetic background including IS elements: IS*26* for *bla*_NDM-5_ and IS*Ecp1* for *bla*_OXA-232_. Commonly, these structures are located on plasmids: IncF, Inc X3, and IncA/C plasmids for *bla*_NDM-5_ and ColE/ColKp3 for *bla*_OXA-232_ [6,40,41,42,45,49].

In addition, the isolates from the ST6260 (ST101-like) clone showed a high diversity in *mgrB* mutations: 10 out of 13 isolates were colistin-resistant, and all of them had *mgrB* variations. Six isolates showed absence of *mgrB* (lack of amplifications), one had a 17 bp deletion, causing a frameshift and premature termination, and in three isolates, aa substitutions were found. In total, 71% (10/14) of colistin-resistant isolates displayed *mgrB* genetic variations. The researchers reported that *mgrB* mutations are the main reason for colistin resistance [50]. In a previous study in Bulgaria, among 27 colistin-resistant isolates, only 9 had *mgrB* variations, but they were members from the ST11 clone carrying *bla*_NDM-1_ [19]. The data from the current study showed a high (71%) percentage of isolates with *mgrB* mutations, especially among ST6260 colistin-resistant isolates (100%). These mutations more probably occurred independently, due to higher colistin usage (almost all patients were treated with colistin). In addition, colistin was used in suboptimal doses (2 × 1 million units, personal information) [20]. The implementation of strict infection control measures in the hospital resulted in the disappearance of colistin-resistant isolates (personal information).

It is worrisome that the investigated isolates were extremely resistant, especially the colistin-resistant members of the ST6260 clone. They were susceptible only to cefiderocol, which could cause serious therapeutic problems. At the end of the study period, we observed three isolates from this clone carrying the same carbapenemases, but they were colistin-susceptible. These facts revealed that isolates from this ST101-like (ST6260) clone showed a great potential to spread in the hospital and to cause therapeutic problems. In addition to cefiderocol, ceftazidime/avibactam in combination with aztreonam could be a good alternative for the treatment of such extremely resistant isolates [51]

Among 13 members of the ST101-like (ST6260) clone, seven isolates were from blood samples and all these patients died (all of them with colistin-resistant variants). Moreover, some previous reports showed that ST101 *K. pneumoniae* can cause bloodstream infections with a high mortality rate [52]. In general, the cause of this increase in mortality is not possible to establish. It could be due to the increased virulence of these isolates, worsening or complications of infection, or underlying comorbidities. The association of ST101 with increased mortality is pointed out by our observation that the investigated in the current study ST101 isolates (colistin-susceptible) has been isolated from patients with unsuccessful therapy (they died). The ST101 isolates produced different carbapenemase KPC-2, in combination with CTX-M-15 ESBL, and we found the FII_AS_ replicon type. Most probably, the transmission of *bla*_KPC-2_ is associated with IncFII plasmids. In the current study, this plasmid type was also detected in the single KPC-2-producing ST258 *K pneumoniae* isolates (colistin-susceptible) and ST16 colistin-resistant isolate. In previous studies in Bulgaria, IncFII plasmids were observed as the main mobile structure that spreads *bla*_KPC-2_ [13,14].

Except the ST101-like (ST6026) clone, among colistin-resistant isolates, we observed three isolates belonging to the ST14 clone. They carried *bla*_OXA-232_ and had intact *mgrB,* which suggests other mechanisms of colistin resistance. PmrA/B, PhoP/Q two-component system, or regulator of PmrAB(Ccr) can influence the negative charge of LPS and thus decrease colistin binding to the LPS [50]. ST14 is a member of ST15 CC and is one of the important international high-risk clones that can carry a wide range of carbapenemases [42]. ST14 carrying *bla*_OXA-232_ has been reported from authors in China [53,54,55]. Interestingly, all three ST14 *K. pneumoniae* isolates were positive for the replicon type ColE, the same as in ST6260, carrying *bla*_OXA-232_ and *bla*_NDM-5_.

The isolates in the current study showed high resistance rates, especially ST6260 members. The colistin- and carbapenem-resistant *bla*_OXA-232_- and *bla*_NDM-5_-positive isolates were susceptible only to cefiderocol, and *bla*_OXA-232_- or *bla*_KPC-2_-positive isolates were susceptible to cefiderocol and ceftazidime/avibactam. The current isolates showed increased resistance to newly introduced antimicrobials such as eravacycline and plazomicin (only one isolate was susceptible). This is in contrast with our previous study on carbapenemase-producing isolates in 2018 showing susceptibility to eravacycline and plazomicin in 76.3% and 90.6% of the isolates, respectively [36]. The higher resistance of the investigated isolates to aminoglycosides, especially ST101-like (ST6026) isolates, can be explained by the presence of *rmtB* 16S rRNA methyltransferases among all except one isolate. In ST14 and ST101 isolates carrying *bla*_OXA-232_ or *bla*_KPC-2_, we observed ArmA methylases (Table 1). The rRNA methyltransferases cause a higher level of resistance to all aminoglycosides, including the recently introduced plazomicin, which increases therapeutic problems with such strains. These isolates were observed in our previous study in 2018 and had a spectrum similar to those common for this period (2017–2018) for Bulgaria (NDM-1 production in 61% and KPC-2 in 23%). Except for plazomicin and eravacycline, a big difference was observed in respect to cefiderocol, if we compare the current study with a previous one (on NDM-1- and KPC-2-producing isolates). Cefiderocol resistance in the previous study was 37.5% according to the EUCAST criteria and 71.8% according to the FDA criteria [36] in comparison with 100% susceptibility in the current study. The method used for susceptibility testing (in our previous study [36]) was the disk diffusion method [36]. In the present study, we used broth the microdilution method with iron-depleted broth. The disk diffusion method was also applied in the current study, and the inhibitory zone diameters around cefiderocol disks were in an Area of Technical Uncertainty (ATU) zone of 21 mm–23 mm [21] for all similar isolates in the previous study [36]. We can suppose that broth microdilution can give more accurate results. The same findings were detected by other authors [56]. Another possibility is different beta-lactamases (carbapenemases NDM-1, AmpC enzyme CMY-4, and extended-spectrum beta-lactamases CTX-M-3/-15) to influence cefiderocol resistance. Some authors associated NDM and AmpC with increased levels of resistance to cefiderocol [57,58].

## 5. Conclusions

This study reports the emergence of extremely resistant *bla*_NDM-5_ and *bla*_OXA-232_ colistin- and carbapenem-non-susceptible *K. pneumoniae* isolates. They were associated with a high *mgrB* variability and a high mortality rate. All *bla*_NDM-5_- and *bla*_OXA-232_-positive isolates belonged to the ST6260 (ST101-like) MLST type. In addition, we detected OXA-232 ST14 isolates and KPC-2-producing ST101, ST16, and ST258 isolates. The colistin- and carbapenem-resistant isolates were susceptible only to cefiderocol for *bla*_NDM-5_- and *bla*_OXA-232_-positive isolates, and to cefiderocol and ceftazidime/avibactam for *bla*_OXA-232_- or *bla*_KPC-2_-positive isolates.

The appearance of extremely resistant *bla*_NDM-5_- or/and *bla*_OXA-232_-positive isolates in the current study leads us to the pre-antibiotic era. The situation has additionally worsened after the COVID-19 pandemic. In Bulgaria, there was a large increase in general antibiotic use, especially for third-generation cephalosporins and carbapenems. The localization of genes encoding carbapenemases on mobile genetic elements (plasmids, transposons, and integrons), which also carry genes for resistance to other antimicrobial agents, can further select problematic microorganisms. The extremely limited choice of appropriate treatment for patients infected with such isolates and their faster distribution imply and highlight the need for urgent measures to control the situation.

## Figures and Tables

**Table 1 antibiotics-13-00677-t001:** Characteristics of patients and *K. pneumoniae* isolates.

No.	Source	Month/ Year	Sex	Age	Main Diagnosis	Treatment Prior Isolation	Ward	Outcome	Bla Genes	ERIC ^1^	ST	Rep ^2^	Met ^3^	MgrB
1K	blood	09/2023	f	69	Kidney insufficiency	amp/sul, cro, mer, col	icu	Ex	OXA-232, NDM-5,	A	6260	colE	rmtB	Δ*mgrB*
2K	blood	09/2023	f	63	Bladder cancer	pip/taz, lnz, col	icu	Ex	OXA-232, NDM-5	A	6260	colE	rmtB	Δ*mgrB*
3K	blood	09/2023	m	82	Bladder cancer	amp/sulb, lev, doxy, col	icu	Ex	OXA-232, CTX-M-15	B	14	colE	armA	WT
4K	urine	10/2023	m	62	Duodenal ulcer	mer	icu	dis	OXA-232, NDM-5	A	6260	colE	rmtB	Δ*mgrB*
5K	abscess	10/2023	m	46	Pancreatic cancer	amp/sul, lev, dox, col	sur	dis	OXA-232 CTX-M-15	B	14	colE	armA	WT
6K	wound	10/2023	m	71	Pancreatic cancer	amp/sul, pip/taz, col	sur	Ex	OXA-232 CTX-M-15	B	14	colE	armA	WT
7K	pleural punctate	12/2023	m	40	Multiple injuries	cfp/sul, col van	icu	dis	OXA-232, NDM-5	A’	6260	colE	rmtB	a124t N42Y **
8K	tracheo- bronchial	12/2023	f	92	Intestinal cancer, Peritonitis	pip/taz, van, col	icu	Ex	OXA-232, NDM-5	A	6260	colE	rmtB	Δ*mgrB*
9K	blood	12/2023	m	74	Peritonitis	pip/taz, lnz	icu	Ex	OXA-232, NDM-5	A	6260	colE	rmtB	Δ72/89 *
10K	blood	01/2024	m	73	Lung cancer	cfp/sul, col	icu	dis	KPC-2 CTX-M-15	D	16	FII	-	WT
11K	drainage fluid	01/2024	m	78	Intestinal cancer	mer, col	icu	Ex	OXA-232, NDM-5	A	6260	colE	rmtB	Δ*mgrB*
12K	blood	01/2024	m	62	Pyelonephritis, Acute kidney injury	cro, col	icu	dis	OXA-232, NDM-5	A’	6260	colE	rmtB	a124t N42Y **
13 K	blood	01/2024	m	81	Pyelonephritis, Septic shock	mer, col	icu	Ex	OXA-232, NDM-5	A’	6260	colE	rmtB	a124t N42Y **
14K	blood	01/2024	m	75	Ca small int IP	mer	icu	Ex	OXA-232, NDM-5	A	6260	colE	rmtB	Δ*mgrB*
15K	blood	01/2024	f	63	Abdominal abscess	cro, col	sur	dis	KPC-2 CTX-M-15	E	258	FII	-	WT
16K	tracheo- bronchial	01/2024	m	91	Pneumonia, Pyelonephritis, Septic shock	cro, col	icu	Ex	KPC-2	C	101	FII	armA	WT
17K	blood	01/2024	m	46	Cirrhosis, Pylonephri-tis, Septic shock, Pneumonia	cro, gen	icu	dis	OXA-232, NDM-5	A’	6260	colE	rmtB	WT
18K	tracheo- bronchial	01/2024	m	75	Mesentery thrombosis, Peritonitis	mer	icu	Ex	KPC-2	C	101	FII	armA	WT
19K	drainage fluid	01/2024	m	50	Peritonitis	mer, col	sur	dis	OXA-232, NDM-5	A	6260	colE	rmtB	WT
20K	drainage fluid	01/2024	f	60	Liver abscess	pip/taz, van, col	sur	dis	OXA-232, NDM-5	A	6260	colE	-	WT

Legend: tracheo-bronch—tracheo-bronchial secretions, G—gender, m—male, f—female, int—intestines, IP—multiple-organ-failure syndrome, icu—intensive care unit, sur—surgery, dis—discharged, Ex—exitus letalis (lethal outcome), cro—ceftriaxone, mer—meropenem, col—colistin, pip/taz—piperacillin/tazobactam, lnz—linezolid, amp/sul—ampicillin/sulbactam, lev—levofloxacin, dox—doxycycline, cip—ciprofloxacin, cfp/sul—cefoperazone/sulbactam, van—vancomycin, gen—gentamicin, ami—amikacin. ^1^, ERIC—Enterobacterial Repetitive Intergenic Consensus; ^2^, rep—replicon typing; ^3^, meth—methylases. WT—wild type with intact *mgrB* gene. * 17 bp deletion 72/89 bp causing frameshift and premature termination; ** asparagin N with tyrosin Y in position 42.

**Table 2 antibiotics-13-00677-t002:** Antimicrobial susceptibility of 20 carbapenem-resistant *K. pneumoniae* isolates.

Antibiotic	S Number (Percent)	I Number (Percent)	R Number (Percent)	Number That Are Susceptible Out of the 14 Colistin-Resistant Isolates	Number That Are Susceptible Out of the 6 Colistin-Susceptible Isolates
Amoxicillin/ clavulanic acid	0 (0%)	0 (0%)	20 (100%)		
Piperacillin/ tazobactam	0 (0%)	0 (0%)	20 (100%)		
Cefuroxime	0 (0%)	0 (0%)	20 (100%)		
Cefoxitin	0 (0%)	0 (0%)	20 (100%)		
Ceftriaxone	0 (0%)	0 (0%)	20 (100%)		
Ceftazidime	0 (0%)	0 (0%)	20 (100%)		
Cefepime	0 (0%)	0 (0%)	20 (100%)		
Imipenem	0 (0%)	3 (15%)	17 (85%)		
Meropenem	0 (0%)	3 (15%)	17 (85%)		
Gentamicin	0 (0%)	0 (0%)	20 (100%)		
Amikacin	1 (5%)	0 (0%)	19 (95%)	1	
Ciprofloxacin	0 (0%)	0 (0%)	20 (100%)		
Levofloxacin	0 (0%)	0 (0%)	20 (100%)		
Fosfomycin	0 (0%)	0 (0%)	20 (100%)		
Trimethoprim/ sulfamethoxazole	4 (20%)	0 (0%)	16 (80%)	1	3
Colistin	6 (30%)	0 (0%)	14 (70%)		6
Ceftazidime/avibactam	7 (35%)	0 (0%)	13 (65%)	4	3
Cefiderocol	20 (100%)	0 (0%)	0	14	6
Eravacycline	3 (15%)	0 (0%)	17 (85%)	1	2
Plazomicin	2 (10%)	0 (0%)	18 (90%)	1	1

Legend: S—susceptible, I—susceptible increased exposure, R—resistant.

## Data Availability

The original contributions presented in this study are included in the article; further inquiries can be directed to the corresponding authors.

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
