# Peer review of "Emergence of blaNDM-5 and blaOXA-232 Positive Colistin- and Carbapenem-Resistant Klebsiella pneumoniae in a Bulgarian Hospital"

_antibiotics, 2024, doi:10.3390/antibiotics13070677_

Round 1

Reviewer 1 Report

Comments and Suggestions for Authors

This is a well-written interesting paper about emergence of NDM Klebsiella pneumoniae in Bulgaria. Since this is an easily-spread pathogen it would be interesting to know the epidemiology in neighbouring countries like Greece and Serbia, and, then, having a  suggestion if the spread comes from south or from north.

Antibiograms presented in the paper suggest that NDM Klebsiella is fully sensitive only to Cefiderocol, while we know that the association of Aztreonam and Avibactam is equally effective even if does not appear from most antibiograms. So, in ICUs is used not only Cefiderecol but even the association Ceftazidime/Avibactam and Aztreonam. A comment about this would be useful for reading physicians.

Please, correct Cefideracol with Cefiderocol throughout the paper (the spelling is correct only in the Tables).

Please improve formatting of Table 2.

Optional references:

Huang YS, Chen PY, Chou PC, Wanh JT: In vitro activities and inoculum effects of Cefiderocol and Aztreonam-Avibactam against metallo-beta-lactamase-producing Enterobacteriaceae. Microbiology Spectrum 2023; 11: e0056923.  doi: 10.1128/spectrum.00569-23

Pragasam AK, Veeraraghavan B, Shankar BA, Bakthavatchalam YD Mathuram A George B, Chacko B, Korula P, Anandan S:  Will ceftazidime/Avibactam plus aztreonam be effective for NDM and Oxa-48-like producing organisms: lessons learnt from in vitro studies. Indian J Med Microbiol 2019; 37: 34-41

Author Response

Dear reviewer,

Thank you very much for the evaluation and the useful comments.  The manuscript is corrected accordingly.

This is a well-written interesting paper about emergence of NDM Klebsiella pneumoniae in Bulgaria. Since this is an easily-spread pathogen it would be interesting to know the epidemiology in neighbouring countries like Greece and Serbia, and, then, having a suggestion if the spread comes from south or from north.

Antibiograms presented in the paper suggest that NDM Klebsiella is fully sensitive only to Cefiderocol, while we know that the association of Aztreonam and Avibactam is equally effective even if does not appear from most antibiograms. So, in ICUs is used not only Cefiderecol but even the association Ceftazidime/Avibactam and Aztreonam. A comment about this would be useful for reading physicians.

Thank you very much. Additional information was included – line 307-309

Please, correct Cefideracol with Cefiderocol throughout the paper (the spelling is correct only in the Tables).

Thank you very much. The corrections were made  throughout the manuscript

Please improve formatting of Table 2.

The Table 2 has been formatted.

Optional references:

Huang YS, Chen PY, Chou PC, Wanh JT: In vitro activities and inoculum effects of Cefiderocol and Aztreonam-Avibactam against metallo-beta-lactamase-producing Enterobacteriaceae. Microbiology Spectrum 2023; 11: e0056923.  doi: 10.1128/spectrum.00569-23

This reference has been included as Ref 51

Reviewer 2 Report

Comments and Suggestions for Authors

This is interesting paper concerning colistin- and carbapenem-resistance Klebsiella pneumoniae in Bulgaria. However, some issues should be improved before acceptance. The main ommision concerns the insufficient information on scientific background of the investigation, provided in Introduction section. Revision is needed in order to clearly explain the role of each monitored gene in antibiotic resistance. This especially concerns methyltransferases genes: some addition in Introduction section should be made, in order to clearly explain the role of 16S rRNA methyltransferases in the colistin resistance. 

Minor changes:

Line 17: "16S RNA methyalses" should be replaced with "16S rRNA methyltransferases"

Line 49: "colistin" should be replaced with "colistin (polymyxin E)"

Lines 51-52: "These modifications occur in pmrCAB and аrnBCADTEF-pmrE operons" should be replaced with "These modifications occur due to mutations in pmrCAB and аrnBCADTEF-pmrE operons" (modificationsd in LPS could be a result of mutations in certain genes, could not occur in operons)

Line 53: "lipopolysaccharides (LPSs)" should be repalced with "LPS" (already introduced abrreviation)

Material and Methods, Paragraph 2.1: The authors should add note concerning ethical approval.

Line 69: repeated word "carbapenem", delete one

Lines 102-105: Unclear definition of XDR (mentioned two times, once inappropriately), some revision is needed.

Line 121: Reference style

Lines 147-148: Please, mention here an abscess isolate among other non-blood sources' isolates.

Line 240: Reference style

Author Response

Dear reviewer,

Thank you very much for the evaluation and the useful comments.  The manuscript is corrected accordingly.

This is interesting paper concerning colistin- and carbapenem-resistance Klebsiella pneumoniae in Bulgaria. However, some issues should be improved before acceptance. The main ommision concerns the insufficient information on scientific background of the investigation, provided in Introduction section. Revision is needed in order to clearly explain the role of each monitored gene in antibiotic resistance. This especially concerns methyltransferases genes: some addition in Introduction section should be made, in order to clearly explain the role of 16S rRNA methyltransferases in the colistin resistance. 

Thank you very much. The additional information has been included in the manuscript, lines – 43-52 and lines 67-80. Additional references was added – N 4-6 and 10-12

Minor changes:

Line 17: "16S RNA methyalses" should be replaced with "16S rRNA methyltransferases"

Corrected throughout the manuscript

Line 49: "colistin" should be replaced with "colistin (polymyxin E)"

Corrected – line  52, 56

Lines 51-52: "These modifications occur in pmrCAB and аrnBCADTEF-pmrE operons" should be replaced with "These modifications occur due to mutations in pmrCAB and аrnBCADTEF-pmrE operons" (modificationsd in LPS could be a result of mutations in certain genes, could not occur in operons)

Thank you very much. Modified – line 59

Line 53: "lipopolysaccharides (LPSs)" should be repalced with "LPS" (already introduced abrreviation)

Corrected – line 60

Material and Methods, Paragraph 2.1: The authors should add note concerning ethical approval.

The current work includes only isolates, obtained during normal diagnostic work. The data from the patients were collected anonymous. The project was approved by the Medical Science Council of Medical University-Sofia. – this statement was add to the manuscript – line 391,392

Line 69: repeated word "carbapenem", delete one

Corrected

Lines 102-105: Unclear definition of XDR (mentioned two times, once inappropriately), some revision is needed.

Corrected – line 124

Line 121: Reference style

 Corrected

Lines 147-148: Please, mention here an abscess isolate among other non-blood sources' isolates.

Inserted – line 168

Line 240: Reference style

Corrected

Reviewer 3 Report

Comments and Suggestions for Authors

This study reported the emergence of 16 NDM-5 and OXA-232 co-producing colistin- and carbapenem-resistant Klebsiella pneumoniae in a Bulgarian hospital. They provided some clinical, phenotypic and genetic analysis of these isolates to highlight the possible clonal shift of carbapenem-resistant Klebsiella pneumoniae in Bulgarian.

1. Although this study is technologically sound, it would be benifit for sequencing the whole genomes of these NDM-5 and OXA-232 co-producing isolates. Because most of these isolates are belonged to ST6260, which might be a potential nosocomial outbreak. This could be determined after the bioinformatics data analysis of the genomes.

2. Table 2, three isolates are intermediate to carbapemems, any possible reasons?

3. What is the breakpoint for determine cefiderocol resistance? What is the possible resistance mechanism of cefiderocol resistance in your isolates?

4. Table 1, I can understand the blaOXA-232 gene is always located on a colE-type plasmid. However, the blaNDM-5 gene is probably not located on the same plasmid. Please verify.

5. Could you determine the genetic surroundings of blaOXA-232 and blaNDM-5 genes?

6. There are four isolates (KP36482022BGR, KP4482023BGR, KP31612023BGR and KP1462023BGR) originated from Bulgaria also co-carrying blaNDM-5 and blaOXA-232 isolates in NCBI database. So this study is not the first one reporting such isolates in Bulgaria.

Comments on the Quality of English Language

The Quality of English Language is overall fine. But there are a few inaccurate expressions. For example, ST types should be STs.

Author Response

Dear reviewer,

Thank you very much for the evaluation and the useful comments.  The manuscript is corrected accordingly.

This study reported the emergence of 16 NDM-5 and OXA-232 co-producing colistin- and carbapenem-resistant Klebsiella pneumoniae in a Bulgarian hospital. They provided some clinical, phenotypic and genetic analysis of these isolates to highlight the possible clonal shift of carbapenem-resistant Klebsiella pneumoniae in Bulgarian.

1. Although this study is technologically sound, it would be benifit for sequencing the whole genomes of these NDM-5 and OXA-232 co-producing isolates. Because most of these isolates are belonged to ST6260, which might be a potential nosocomial outbreak. This could be determined after the bioinformatics data analysis of the genomes.

We appreciate the reviewer’s opinion and absolutely agree. We will soon start the project to perform WGS on some of NDM-5 and OXA-232 isolates, but unfortunately at present, for financial reasons, we cannot perform WGS.

2. Table 2, three isolates are intermediate to carbapemems, any possible reasons?

These isolates were positive for blaOXA-232. In general, the producers of OXA-48 and its variants have low MICs of carbapenems, and OXA-232 have decreased carbapenem MICs in comparison with OXA181 and OXA-48.  The addition was made in line – line 255, 256

3. What is the breakpoint for determine cefiderocol resistance? What is the possible resistance mechanism of cefiderocol resistance in your isolates?   

The cefiderocol resistance breakpoints used in this manuscript were according EUCAST (S <2mg/L and R>2mg/L). All the isolates in this study (MIC determination) were susceptible.

            In the previous study, cefiderocol susceptibility was tested with the disk method, and we observed many isolates in ATU zone (EUCAST,2023), which we considered as resistant. We revised the paragraph to be clearer.line 347-349  One possibility for the high cefideracol resistance in the previous study  is a methodology problem (the previous isolates were tested only with disk diffusion method, while some authors suggest that MIC determination with iron depleted agar gives more precious results.

 In general, cefiderocol resistance can be associated with the type of carbapenemases – NDM producers showed much higher cefiderocol nonsusceptibility (42–59% in some reports [Longshaw et al, 2020) of in NDM-producing clinical isolates. Our isolates (from the previous study, which showed a high cefiderocol resistance) were NDM-1 producers, most of them in combination with CMY-4 and CTX-M-3 or -15. There is evidence that some AmpC enzymes also contribute to cefideracole resistance (Shields et al, 2020) Among OXA-type β-lactamases, only OXA-427 was correlated to cefiderocol resistance (Jacob et al, 2021)

In addition, some mutations in siderophore receptors CirA and/or Fiu in enterobacteria can also cause cefideracol resistance [Karakonstantis S, 2022].

Longshaw C., Manissero D., Tsuji M., Echols R., Yamano Y. In vitro activity of the siderophore cephalosporin, cefiderocol, against molecularly characterized, carbapenem-non-susceptible Gram-negative bacteria from Europe. JAC-Antimicrob. Resist. 2020;2:dlaa060.

Shields R.K., Iovleva A., Kline E.G., Kawai A., McElheny C.L., Doi Y. Clinical Evolution of AmpC-Mediated Ceftazidime-Avibactam and Cefiderocol Resistance in Enterobacter cloacae Complex Following Exposure to Cefepime. Clin. Infect. Dis. 2020;71:2713–2716.

Jacob A.-S., Chong G.-L., Lagrou K., Depypere M., Desmet S. No in vitro activity of cefiderocol against OXA-427-producing Enterobacterales. J. Antimicrob. Chemother. 2021;76:3317–3318. doi: 10.1093

Karakonstantis S, Rousaki M, Kritsotakis EI. Cefiderocol: Systematic Review of Mechanisms of Resistance, Heteroresistance and In Vivo Emergence of Resistance. Antibiotics (Basel). 2022;11(6):723. Published 2022 May 27.

The insertion was made in the text – line 360-363

4. Table 1, I can understand the blaOXA-232 gene is always located on a colE-type plasmid. However, the blaNDM-5 gene is probably not located on the same plasmid. Please verify.

All our isolates producing NDM-5 also produced OXA-232. There were not isolates in the current study producing only NDM-5 carbapenemase. However, we have isolates producing only OXA-232, which showed colE-type plasmid.  blaNDM-5 is associated with distribution of IncA/C, IncF, IncX3 [ Kopotsa, K.].  blaOXA-232 can be found in nonconjugative colE, colKP-3 plasmids [Kopotsa, K et al, 2019; Chen et al, 2023]

Kopotsa, K., Osei Sekyere, J. & Mbelle, N. M. Plasmid evolution in carbapenemase-producing Enterobacteriaceae: A review. Ann. N Y Acad. Sci. 2019;1457(1):61-91.

Chen T, Xu H, Chen Y, et al. Identification and Characterization of OXA-232-Producing Sequence Type 231 Multidrug Resistant Klebsiella pneumoniae Strains Causing Bloodstream Infections in China. Microbiol Spectr. Published online March 22, 2023. doi:10.1128/spectrum.02607-22

The additional information was included – line 277-280

  1. Could you determine the genetic surroundings of blaOXA-232 and blaNDM-5 genes?

In the transposition of the blaNDM-5 gene, IS elements such as IS26, ISAba125 are most often involved (Takei, Nepal).  This structure can be found on IncF, Inc X3 and IncA/C plasmids.
blaOXA-232  is associated with ISEcp1 which is almost entirely deleted [Potron,2013]. This structure is located on ColE/ColKp3 plasmids.

The additional information was included – line 281-286

  1. There are four isolates (KP36482022BGR, KP4482023BGR, KP31612023BGR and KP1462023BGR) originated from Bulgaria also co-carrying blaNDM-5 and blaOXA-232 isolates in NCBI database. So this study is not the first one reporting such isolates in Bulgaria.

The correction was made, the statements “first report” were removed .

The Quality of English Language is overall fine. But there are a few inaccurate expressions. For example, ST types should be STs.

Corrected – line 202, 264

Round 2

Reviewer 2 Report

Comments and Suggestions for Authors

The manuscript has been upgraded suficiently to be accepted in the present form

Reviewer 3 Report

Comments and Suggestions for Authors

All comments have been addressed.